# Epidemic-induced local awareness behavior inferred from surveys and genetic sequence data

Gergely Ódor [1,2,3] ✉ & Márton Karsai [1,2]

Behavior-disease models suggest that pandemics can be contained cost-effectively if individuals take preventive actions when disease prevalence rises among their close contacts. However, assessing local awareness behavior in real-world datasets remains a challenge. Through the analysis of mutation patterns in clinical genetic sequence data, we propose an efficient approach to quantify the impact of local awareness by identifying superspreading events and assigning containment scores to them. We validate the proposed containment score as a proxy for local awareness in simulation experiments, and find that it was correlated positively with policy stringency during the COVID-19 pandemic. Finally, we observe a temporary drop in the containment score during the Omicron wave in the United Kingdom, matching a survey experiment we carried out in Hungary during the corresponding period of the pandemic. Our findings bring important insight into the field of awareness modeling through the analysis of large-scale genetic sequence data, one of the most promising data sources in epidemics research.

The COVID-19 pandemic has highlighted several pivotal shortcomings that demand comprehensive examination within our society[1]. One of the most important lessons was the need for more effective social interventions, which can ensure the adherence to the necessary containment measures during future pandemics[1,2]. Manifesting as a social dilemma, restrictive measures generate a conflict between long-term collective interest and short-term self-interest[3], and it can be difficult to convince individuals to cooperate, especially if the cooperative behavior needs to be sustained for longer time periods[4–6]. Among interventions that raise awareness and promote cooperative behavior, a combination of community engagement, accurate monitoring, and transparent reporting of the impact of restrictions has been found the most consistently effective approach[7,8].

Recognizing the importance of the problem, the research community responded to the emergence of the COVID-19 pandemic by closely monitoring and actively reporting the changes in epidemic awareness[9,10]. However, most of these studies focused on *global awareness*, defined as changes in preventive behavior based on publicly available information[11,12], such as global case-counts or governmental restrictions. In contrast, *local awareness* is defined as changes in preventive behavior driven by locally available information about disease prevalence or locally spreading beliefs unrelated to disease dynamics[11,12]. Among prevalence-based local awareness mechanisms, in this paper we are primarily interested in voluntary behavioral changes, motivated by concerns for one's own health or the health of others[13], instead of the behavioral changes enforced by public health authorities based on local contact tracing[14]. Substantial model-based evidence suggests that voluntary, prevalence-based local awareness can effectively reduce the pandemic threshold and the size of the epidemic[12,15–17]. Intuitively, since local awareness relies on local information, it may serve as a targeted and more efficient method to control the epidemic compared to its global counterpart. Despite its potential, the limited data availability on individual-level disease prevalence and voluntary preventive behaviors makes local awareness more challenging to monitor at a large scale, leaving a significant gap in our understanding of its impact in real scenarios.

[1]Department of Network and Data Science, Central European University, Vienna, Austria. [2]National Laboratory of Health Security, HUN-REN Alfréd Rényi Institute of Mathematics, Budapest, Hungary. [3]Institute for Hygiene and Applied Immunology, Center of Pathophysiology, Infectiology and Immunology, Medical University of Vienna, Vienna, Austria. ✉e-mail: gergelyodor.research@gmail.com

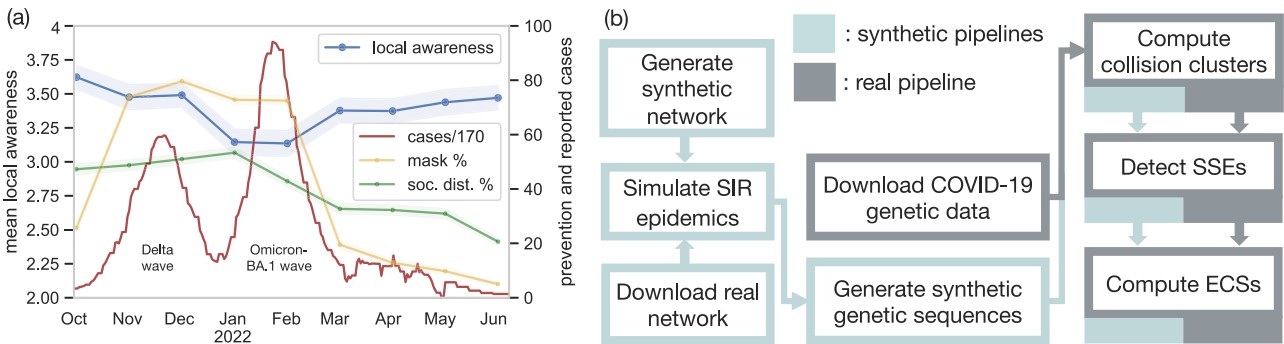

**Fig. 1 | Local awareness behavior inferred from surveys and genetic sequence data. a** The MASZK Hungarian telephone survey, with 1000 participants in each of the 9 months, shows that the mean local awareness score (in blue) remains relatively constant throughout the recording period, except during the Omicron wave, when the score drops. The government-imposed preventive measures (mask wearing, in yellow, and social distancing, in green) show a different temporal pattern. For all survey results, we show the mean response for each

month, with confidence intervals calculated under the assumption of normality. The daily number of cases (with a rolling-mean of 7 days, normalized by 170) are shown in red. Source data are available in Supplementary Data 1. **b** Our proposed pipelines to generate synthetic (blue) and process real genetic sequence data (gray) to compute collision clusters, superspreading events (SSEs), and finally Event Containment Scores (ECSs) – a proxy measure for local awareness behavior.

To fill the gap in monitoring voluntary, prevalence-based local awareness behavior, we conducted a representative telephone survey asking 9000 participants over 9 months during the Delta and the Omicron waves in Hungary as part of the MASZK national survey[18]. The responders were asked to rate their willingness to undertake stricter preventive measures (such as increased mask wearing or social distancing) if the prevalence of the disease increased among their close contacts. The survey results show an unexpected pattern (Fig. 1a). While the measured local awareness scores stayed relatively constant throughout the collection period, including the Delta wave of the pandemic, we observed a drop in local awareness during the Omicron BA.1 wave, which rebounded promptly after the wave has ended.

The measured local awareness scores show a distinctive temporal pattern compared to the standard protective measures, which we also assessed in the same survey. Fig. 1a shows that mask wearing stayed constant throughout both the Delta and the Omicron waves, while social distancing dropped during the Omicron wave, but did not rebound after the wave has ended. These additional survey results also rule out the hypotheses that the drop in local awareness scores can be explained exclusively by the responders inability to perform stricter measures during the Omicron wave, or by the relatively lower risk of hospitalization and death posed by the Omicron variant.

According to our interpretation, the observed drop in local awareness scores can be attributed to a form of pandemic fatigue[4,5]; a decrease in voluntary preventive behavior due to the complex interplay of various psychological factors. However, since the general adherence to regulations showed a very different pattern compared to the local awareness behavior in Fig. 1a, the observed "local-awareness fatigue" is likely to have a very different psychological explanation, which our survey was not designed to reveal. Instead of speculating about the mechanisms of the observed phenomenon, we focus on two important questions about the impact of our finding: (i) do other countries show similar changes in local awareness behavior? (ii) does the observed drop in self-reported local awareness have a measurable impact on the spread of the epidemic? To answer these questions we turn to the analysis large-scale genetic sequence data, which contains hidden, but accessible information about the local spread of the epidemic.

While genetic data raises relatively minor privacy concerns[19], and it is unparalleled in terms of availability at the individual level, inferring behavioral information from genetic sequences is a challenging task. In phylodynamics[20,21], human behavior is typically estimated based on the phylogenetic tree reconstructed from the observed sequences[22]. However, current tree reconstruction methods have a number of

limitations. First, traditional methods are computationally intensive and it is difficult to scale them to datasets with more than a few thousand sequences[23,24]. Since the COVID-19 pandemic, there has been significant progress in developing more scalable methods[25], and releasing publicly available trees for further analysis[26,27]. However, processing millions of SARS-CoV-2 genetic sequences remains a challenge[28], and the publicly shared pre-computed trees do not have the same coverage as the Global Initiative on Sharing All Influenza Data (GISAID) dataset, which contains over 16 million SARS-CoV-2 genetic sequences, with a 5-15% sequencing rate in several countries[29]. Second, working with general-purpose methods or highly pre-processed datasets can significantly lower the statistical power of our results, especially since previous methods were not optimized to measure local awareness behavior. Instead, we process this new dataset of unprecedented size by focusing on a simple and tractable statistic that does not require the reconstruction of the phylogenetic tree – the size distribution of the clusters of identical genetic sequences over time. Similar tree-free methods with different applications have been recently proposed by[30–33]. In essence, we break up the global epidemic into thousands of sub-epidemics with identical genetic code to infer patterns of local awareness. Since each sub-epidemic contains only very noisy information about general local awareness patterns in the population, we focus on one of the most robust features of the dataset: *superspreading events*.

The role of superspreading events as the driving force of the COVID-19 pandemic was well-established in early 2020[34]. Since then, there has been a remarkable research effort to understand the potential of targeted interventions to prevent or contain superspreading events[35–37], and to document the effect of these interventions in case studies based on contract tracing[38,39]. It has also been shown via phylogenetic analysis that superspreading events may have vastly different downstream infection patterns – some are contained very quickly, while others lead to sustained community transmission[40].

Although the determinants of the outcomes of superspreading events is are still an active research area, in this paper we hypothesize that a quickly contained events are a sign of local awareness behavior. Based on this hypothesis, we propose to infer local awareness behavior exclusively from local spreading patterns, as we do not know the local information that was available to the individuals who were sampled in the genetic sequence dataset. Since our hypothesis may not hold for each individual event, we aggregate the outcome of hundreds of events throughout the entire dataset, and we also employ a number of validation steps on the resulting signal based on various exogenous variables and simulation experiments. We note that we are not able to

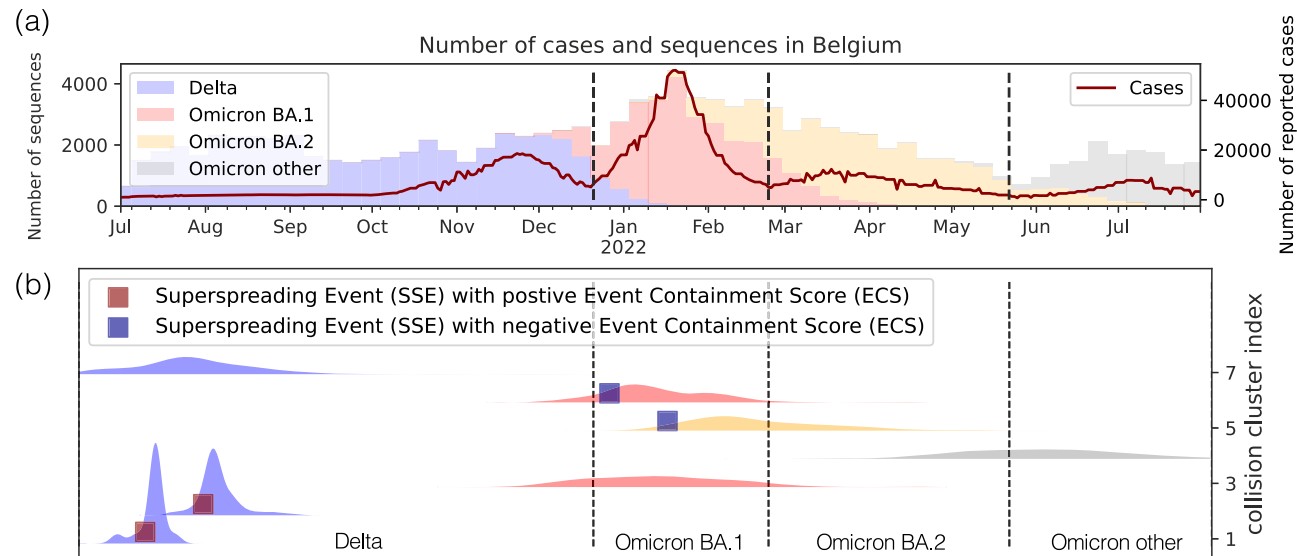

**Fig. 2 | Weekly number of SARS-CoV-2 genetic sequences shared in Belgium (2021-2022). a** Bar plot showing the weekly number of SARS-CoV-2 genetic sequences collected in Belgium and shared via the GISAID platform for each major variant from July 2021 to July 2022. Dashed lines indicate the weeks when a new variant became dominant. The solid red line represents the number of reported SARS-CoV-2 cases. **b** Temporal evolution of seven identified collision clusters in Belgium. Within these clusters, our proposed thresholding approach detected four superspreading events, marked with square symbols – typically occurring near the beginning of a cluster. The color of each square represents the sign of the corresponding containment score.

distinguish voluntary and externally-imposed local awareness based on our main hypothesis – a limitation, which we address in the Discussion.

The rest of the paper is organized as follows. First, we develop a pipeline to detect superspreading events based on the size distribution of clusters of identical genetic sequences, and to measure the resulting secondary infections by assigning each superspreading event an Event Containment Score (ECS, see Fig. 1b). Intuitively, ECS is a proxy for the level of adaptive local awareness behavior, which we confirm via extensive simulation results on synthetic epidemic models with local awareness. In the GISAID dataset, we demonstrate that the ECS correlates positively with the Oxford Containment Health Indices[41] in European countries, but not with potential confounders, such as the sequencing rate or the attack rate. Finally, we show that – similarly to the Hungarian survey – there was a drop in the ECS scores in the United Kingdom during the Omicron BA.1 wave. In addition to providing evidence for the impact of local awareness in multiple countries, our methods pave the way for future interdisciplinary studies that monitor behavioral patterns using large-scale genetic sequence data.

## Results

### Method overview

Our analysis is based on the detection of superspreading events and the assignment of containment scores to each event by quantifying secondary infections (Fig. 1b). As the first step of the pipeline, we download and preprocess the GISAID EpiCoV database[29]. Unfortunately, the sequencing rate in Hungary was too low for a meaningful comparison with the survey results. In the interest of data quality and a close match with the survey experiment, we focused on sequences collected in European countries with a sequencing rate of at least 2% from the Delta, Omicron BA.1 and BA.2 variants. For our analysis, we mainly relied on the amino-acid-level substitution dataset precomputed from the raw clinical genetic sequences by the GISAID pipeline – a dataset that has been previously used to detect variants of interest[30] and to visualize mutation trends[42]. We partition the genetic sequences with identical amino acid substitutions into subsets, which we call *collision clusters* (CCs). We group together collision clusters that were collected in the same country and that belong to the same

variant, filtering out clusters that are prevalent in multiple countries. Following[43], we assume that SARS-CoV-2 viruses from the same variant had similar fitness profiles, there was no significant selection between them, and the infection probability and recovery time of the patients were similar.

We detect superspreading events in each collision cluster by tracking unexpectedly large increases in their size after proper normalization (see Methods). Our superspreading event detection method is closely related to previous thresholding approaches[33,40], requires only minor preprocessing. The detected events agree with our intuition after visual inspection (Fig. 2b) and a more in-depth analysis based on location metadata in Supplementary Section A. Thereafter, we assign Event Containment Scores (ECSs) to each superspreading event by comparing the size of the collision clusters after superspreading events and after appropriately selected baseline events during the same time period (see Methods). Finally, to acquire aggregate descriptions of event containment, we compute the median of ECS values in each country-variant pair $c$, denoted by MECS; the output of the pipeline in Fig. 1b. Intuitively, a positive MECS means that superspreading events typically led to smaller collision cluster sizes, and therefore fewer secondary infections than the baselines, i.e. the superspreading events were well-contained (Fig. 2b, red squares). Similarly, a negative ECS would suggest superspreading events that were not contained as well as the baselines (Fig. 2b, blue squares).

Both the superspreading event detection and the ECS assignment algorithms are efficient but imperfect methods, potentially introducing significant amounts of noise in our results. However, we expect that if enough superspreading events are detected in a country-variant pair, the median of the ECS values will still contain information about event containment, and subsequently, local awareness behavior. We confirm this hypothesis by simulation results and by the analysis of COVID-19 genetic sequences.

### Event Containment Scores on Synthetic Genetic Sequence Data

We set up a synthetic pipeline (Fig. 1b) to generate genetic sequence datasets similar to the GISAID EpiCoV dataset, which we can analyze with our superspreading event detection and ECS assignment pipeline. First, we simulate Susceptible-Infected-Recovered (SIR) epidemics on

various synthetic and real networks, then we apply the Jukes-Cantor[44] genetic substitution model on the resulting infection tree to produce genetic sequence data (see Methods). To model the combined effect of not all infectious individuals being identified (detection rate), and not all identified individuals being sequenced (sequencing rate), we randomly subsample the generated sequences with probability $p$. Finally, we compute the MECS values as before, with $c$ denoting the model parameters instead of the country-variant pair.

For the underlying network, we select four real social networks and three types of synthetic random networks. Two company friendship networks[45], that encode personal connections (recorded by Facebook), have medium size (around 5000 nodes), and have similar characteristics as the contact networks on which a viral disease (such as SARS-CoV-2) can spread. Two online social networks, the Google+ friendship network[46], and the Twitter mutual mention network[47] are large (over 200,000 nodes), and they model the underlying network of online contagion processes (e.g., rumor, misinformation). All 4 networks have a heterogeneous degree distribution and a relatively high clustering coefficient (Supplementary Fig. B8). To model these characteristics separately, we select three synthetic network models: the Configuration Model has a heterogeneous degree distribution but no clustering, the Stochastic Block Model (SBM) has high clustering but a homogeneous degree distribution, and the Geometric Inhomogeneous Random Graph (GIRG) model[48], which has both a heterogeneous degree distribution and high clustering. On all network models, due to the heterogeneous degree distribution (or the community structure in case of the SBM), we expect large infection events that can be detected with our superspreading event detection algorithm.

We include local and global awareness in our simulations as a modification of the SIR model with adaptively changing infection probabilities. Inspired by[49], for local awareness we set the infection probability of an infectious node $u$ at time $t$ to be

$$\beta_{u,t} = \beta_0 e^{-\alpha_l I_{u,t}}, \qquad (1)$$

where $\beta_0 \in [0, 1]$ is the basic infection probability, $\alpha_l$ sets the strength of the local awareness behavior, and $I_{u,t}$ is the number of infectious neighbors of node $u$ at time $t$. In case of global awareness, all infectious nodes $u$ have the same infection probability at time $t$:

$$\beta_{u,t} = \beta_0 e^{-\alpha_g I_t/N}, \qquad (2)$$

where $I_t$ is the total number of infectious nodes in the network, $\alpha_g$ sets the strength of the global awareness behavior, and $N$ is the size of the network. The exponential function in equation (1) (resp., (2)) aims to model a scenario where each neighbor (resp., node) may alert node $u$ about their infectious status, and each of these independent alerts cause a multiplicative reduction in the infection probability. This model is similar to alternative approaches that treat local awareness as a contagion process, where the probability of staying unaware decays exponentially in the number of aware neighbors[12,15,16]. As a robustness check, we also implement linearly decaying local awareness functions, since it has been reported that they may be more cost-effective based on an epi-economic point of view[50] (Supplementary Fig. B7).

In Fig. 3, we plot the dependence of MECS on the awareness-strength parameters $\alpha_l$ and $\alpha_g$ and two potential confounding factors: the basic infection probability $\beta_0$, and the subsampling probability $p$. The results indicate that MECS primarily depends on the parameter $\alpha_l$ (Fig. 3a). Importantly, we were only able to generate positive MECS values with the local awareness model, apart from the noisy MECS values near zero for low subsampling probability in smaller networks. This is a strong indication that the positive MECS values are signs of local awareness behavior.

The observation that only local awareness can produce positive MECS values has an intuitive explanation. When a superspreading event occurs, there is usually a common trait between the individuals that become infected at the same time; they all tend belong to the same community as the initial infector. It is also likely that there exist many additional individuals who belong to the same community, but do not become immediately infected. Indeed, reports of early

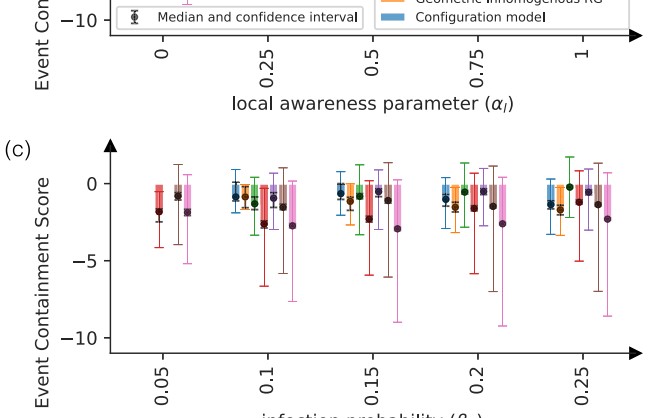

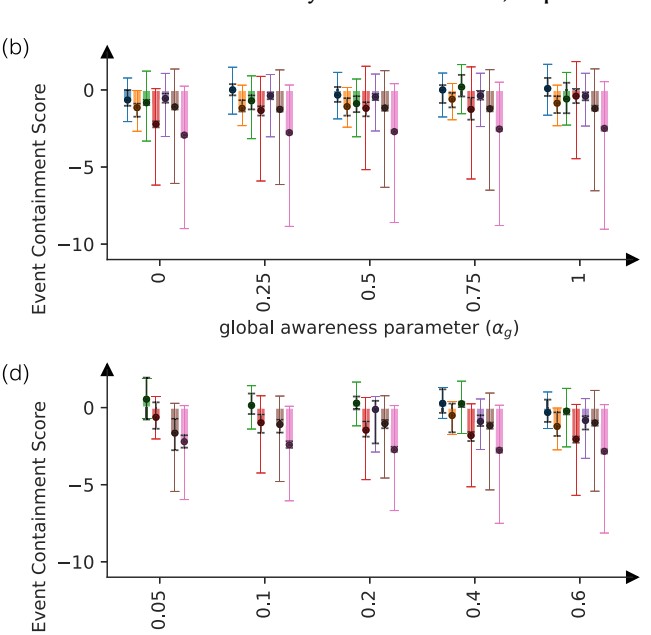

**Fig. 3 | Event Containment Scores (ECS) and their median values computed in simulated genetic data.** Epidemics were simulated on synthetic and real networks as a function of **a** the local, **b** the global awareness function parameter, **c** the infection probability and **d** the subsampling probability of the resulting genetic sequences. For each set of parameters, we simulated $n = 200$ independent epidemic processes with different random seeds. Colored intervals show the 25th and 75th percentiles of the ECS values, while black intervals indicate confidence intervals for the median, computed using a normal approximation. Source data are available in Supplementary Data 2. When not stated otherwise, all parameters are set to be their default values $\alpha_l = 0$, $\alpha_g = 0$, $\beta_0 = 0.15$, and $p = 1$. We observe positive Median Event Containment Scores (MECS) in the case of local awareness, and noisy MECS values near zero if the subsampling probability is low.

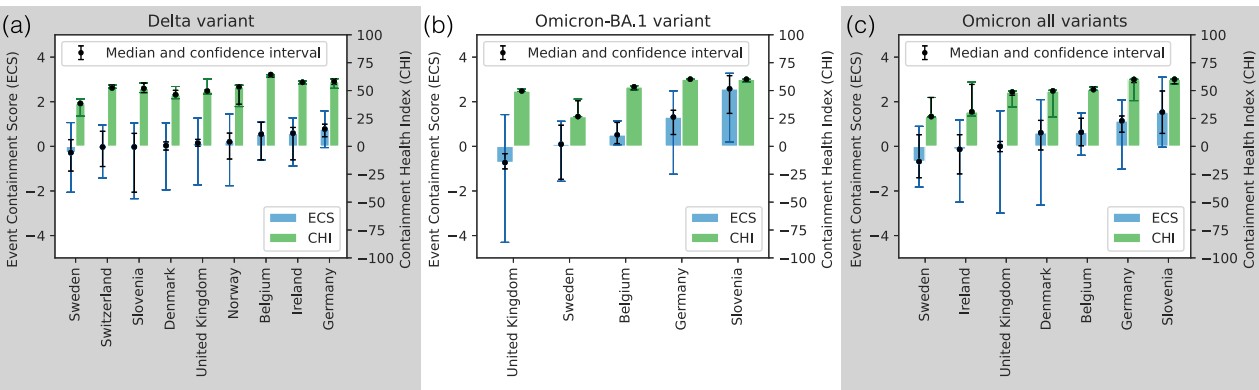

**Fig. 4 | Event Containment Scores (ECS) and Containment Health Index (CHI) in real data.** Bar plots and black dots mark median ECS (blue) and CHI (green) values in European countries with at least 15 detected superspreading events in the (**a**) Delta, **b** Omicron BA.1 variants, and **c** when all Omicron variants are merged. The number of ECS values corresponding to each Median ECS (MECS) value is shown in Supplementary Table B.1. Colored intervals show the 25th and 75th percentiles of the distribution, while black intervals indicate confidence intervals for the median, computed using a normal approximation. Country-variant pairs with a confidence interval larger than 3 around the MECS values are filtered out. Gray background signifies a statistically significant correlation between MECS and the median CHI values (Table 1).

superspreading events during COVID-19 do not report *all* individuals becoming infected in the communities at the same time[51,52], and the same is true in simulations, unless the infection probability inside the community is close to 1. If the structure of the contact network remains unchanged after the superspreading event, then these additional community members become infected in the next timestep (week), which causes the number of sequences in the collision cluster to grow, and therefore produces a negative MECS value. Note that there are extreme examples of static networks and epidemic parameters that produce a positive MECS value. For instance, in a star network with infection probability close to 1, an epidemic from the center node produces a single superspreading event, and then dies out in the next step, resulting in MECS > 0. However, we conclude that besides a few extreme cases, positive MECS values, such as the ones observed in the empirical dataset in Fig. 4 – are signs of local awareness behavior.

### Local awareness in the COVID-19 Genetic Dataset – Spatial analysis

We compute the MECS values for all country-variant pairs with at least 15 detected superspreading events during the Delta or the Omicron BA.1 variants in the GISAID EpiCoV dataset (Fig. 4a, b), and we analyze how these values are related to behavioral metrics and potential confounding factors. Since we only have 5 datapoints in the Omicron BA.1 wave due to data availability, we also performed the same experiment on all Omicron sequences merged together in Fig. 4c).

Fig. 4a, b shows statistically significantly positive containment scores for Germany in the Delta wave and Germany, Slovenia and Belgium during the Omicron BA.1 wave – a sign of local awareness behavior established in the previous section. To understand the

factors that could explain the variability between the observed MECS values, we compute the sequencing rate, the attack rate, and the Containment Health Index (CHI) in each country-variant pair (see Methods). CHI is a composite epidemic response measure based on thirteen policy indicators maintained by the Oxford Coronavirus Government Response Tracker (OxCGRT) project, similarly to the stringency index[41]. We plot the CHI in Fig. 4a–c, and we compute the Spearman-r statistic between them and the MECS values (Table 1). Interestingly, we find a positive correlation between the MECS values and the Containment Health Index, which becomes statistically significant in the Delta wave and when we merge Omicron waves into a single dataset, suggesting that government policies may also impact the local awareness behavior we measure.

While we find no significant correlation between the MECS values and the attack rate (Supplementary Fig. B3), we do observe a statistically significant negative correlation with the sequencing rate during the Delta wave (Table 1), which could suggest that MECS is an artefact of how the data was collected. However, in the Delta wave, sequencing rate and CHI happened to be highly and negatively correlated, potentially because countries aimed to lift the economic burden of strict containment policies by a higher quality sequencing and monitoring project. In the Omicron BA.1 wave and when all Omicron samples are merged, there is no significant correlation between the MECS values and the sequencing rate, suggesting that MECS measures a behavioral signal instead of confounding effects.

### Local awareness in the COVID-19 Genetic Dataset – Temporal analysis

Having validated containment scores in real and synthetic datasets, we return to our motivating research question; whether drops in local awareness behavior can be observed in the genetic sequence dataset during the Omicron BA.1 wave of the COVID-19 pandemic. One approach to answer this question is to compare the variant-aggregated MECS scores from Fig. 4 between the Delta and the Omicron BA.1 waves. Fig. 5a shows that MECS values during the Omicron BA.1 wave were lower compared to the Delta wave in Ireland and the United Kingdom, with other European countries either showing no change between the two waves (Belgium), or an increased MECS in the Omicron BA.1 wave (Sweden, Denmark, Germany, Slovenia). As opposed to the spatial analysis in Fig. 4, the temporal trends in the MECS do not seem to be explained by the Containment Health Index. Fig. 5b shows that while the ranking of the MECS values and the CHI are still correlated, the median stringency of the policies became more relaxed only

**Table 1 | Spearman rank correlation coefficients (Spearman's ρ) and corresponding two-sided p-values were computed between MECS values and the exogenous variables (Containment Health Index plotted in Fig. 4a–c and sequencing rate plotted in Supplementary Fig. B3)**

|  | Containment Health Index | | | sequencing rate | | |
|---|---|---|---|---|---|---|
|  | Delta (a) | BA.1 (b) | Omicron all (c) | Delta (a) | BA.1 (b) | Omicron all |
| Spearman's ρ | 0.800 | 0.800 | 1.000 | −0.867 | 0.000 | −0.107 |
| p-value | 0.010* | 0.104 | 0.000* | 0.002* | 1.000 | 0.819 |

No correction was applied for multiple comparisons. Significant p-values (p < 0.05) are indicated with an asterisk.

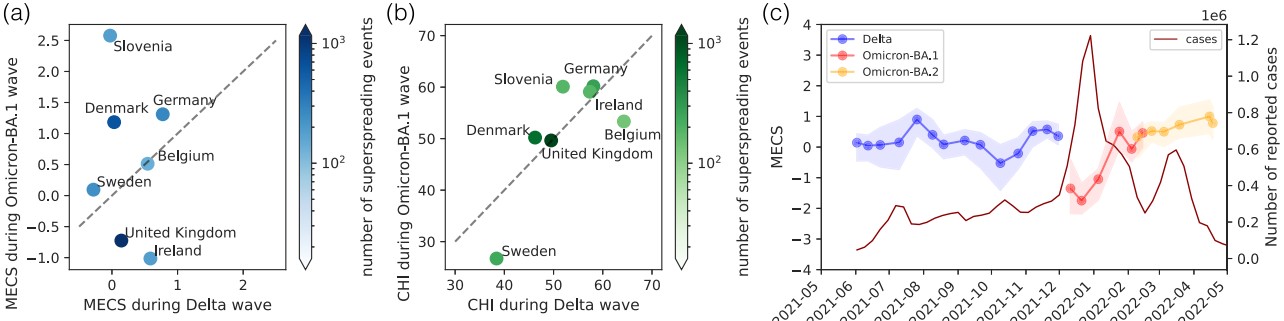

**Fig. 5 | Comparison of event containment during the Delta and the Omicron BA.1 variants. a** Median Event Containment Scores (MECS) during the Delta and the Omicron BA.1 variants as computed in Fig. 4. Datapoints below the dashed ($x = y$) line hint at drops in local awareness during Omicron BA.1 variant. **b** Containment Health Index (CHI) during the Delta and the Omicron BA.1 variants as computed in Fig. 4. **c** MECS values computed biweekly with a 4-week sliding window in the UK for the Delta, Omicron BA.1 and BA.2 variants. Confidence intervals were computed using a normal approximation, and datapoints with a confidence interval larger than 2 are filtered out. We observe a drop in MECS in December 2021 - January 2022 during the Omicron BA.1 wave.

in Sweden and in Belgium, and no change can be observed in the case of Ireland and the UK. However, the purpose of ECS values is to measure the impact of local awareness instead of the policy stringency in the country. As an alternative explanation, we highlight the fact that the Omicron BA.1 wave arrived in the UK and to Ireland a few weeks before its arrival to continental Europe, during the late December instead of early January. The extreme changes in mixing behavior during the holiday season may have contributed to the lower containment scores measured in Fig. 5b.

Up until this point, we focused on the MECS values, computed as the median of all ECS values for a country-variant pair. However, in the United Kingdom – where thousands of superspreading events are detected across multiple variants – a higher temporal resolution can be achieved by calculating the median of ECS values biweekly with a 4-week sliding window. The resulting signal (Fig. 5c), obtained purely based on genetic sequence data, shares a remarkable similarly with the Hungarian survey results in Fig. 1a. Both curves show a relatively stable signal between October 2021 and July 2022, with a smaller drop during November 2021 and a significant drop at the beginning of the Omicron BA.1 wave.

Notably, the decline in ECS values coincides with the transition from the Delta to the Omicron BA.1 wave, raising concerns that this trend may reflect the increased transmissibility of the BA.1 variant compared to Delta. However, simulations (Fig. 3c) suggest that transmissibility alone has only a minimal impact on ECS values. Furthermore, the BA.2 variant, which also had increased transmissibility relative to BA.1, does not exhibit a similar discontinuity in the ECS signal. A possible alternative explanation is that a temporal drop in ECS could also be driven by spatial variations in behavior. Although our analysis treats the UK as one homogeneous population, it has been reported that the introduction of the BA.1 variant into the UK was initially localized in the London area[53], and the drop in ECS could be a result of a limited ability to engage in awareness behavior in this region. In contrast, the Hungarian dataset was a representative survey, and with the Omicron wave arriving later in Hungary, the introduction of the disease was likely more uniform than in the UK.

While the uncertainties and the differences in the data collection render the direct comparison of the British ECS signal and the Hungarian survey inherently challenging, their alignment opens an array of new questions and research directions in behavioral epidemiology. Moreover, the temporal resolution of the ECS signal in the UK underscores the potential of our approach as a new tool to evaluate the impact of local awareness behavior during a pandemic situation.

## Discussion

In epidemic surveillance, there is usually a trade-off between the breadth and the depth of the data we can access. On one end, we have aggregate

case counts, that give a macroscopic view on the epidemic; one the other end we have a handful of case-studies, which tell about the local spread. Survey results provide a representative depiction of self-reported human behavior, however, they lack sufficient information on disease spread to support conclusions beyond forming hypotheses.

In this paper, we observe local awareness behavior in two complementary datasets: a Hungarian survey dataset and the dataset of clinical genetic sequences collected during the COVID-19 pandemic. We first show that the survey results indicate a drop in local awareness behavior during the Omicron wave of the COVID-19 pandemic. Based on the survey results, we formulate a question, whether this drop occurred and caused noticeable changes in the spread of the disease in other countries as well. To address this question, we introduce a methodology that utilizes genetic sequence data, striking a new balance between micro and macroscopic epidemic surveillance.

As with any trade-off, our proposed analysis comes with a number of limitations. We identify superspreading events based on simple thresholding of sequence counts, which is less accurate than manual contact tracing, where more metadata and more context about infection events can be taken into account. Consequently, we only compute highly aggregated statistics on the detected events. One ECS gives only very noisy information about the outcome of each superspreading event, and only the median of all ECSs, the MECS value has the statistical power to say anything about local awareness in region $c$. Since the number of genetic sequences we have available since COVID-19 is unprecedented, and the new tools to analyze it are just being developed[23], our results too have to be confirmed by further research.

Besides the inherent noise in the analysis, our results rely on the hypothesis that quickly contained superspreading events are a sign of local awareness behavior. While we validate this hypothesis in standard epidemic models, the true determinants of the outcomes of superspreading events are still an active area of research[40]. Furthermore, our approach is not able to distinguish voluntary or externally-imposed local awareness behavior. This limitation is alleviated by the fact, that during the Delta and the Omicron waves the contact tracing efforts in many European countries were overpowered by the number of cases in the population, suggesting that most of the measured signal is due to voluntary local awareness.

In addition to recognizing the inherent limitations of the methods, it is crucial to interpret the comparison of ECS scores and survey results carefully. Figs. 1a and 5c reveal a strikingly similar pattern; however, the former captures self-reported willingness to adopt stricter protective behaviors, while the latter reflects the observed effects of local awareness behavior. The observed drops during the Omicron BA.1 wave may have distinct underlying causes in the two countries, with the former potentially being influenced by

psychological factors and the latter by the seasonality of population mixing patterns.

Despite these limitations, the new methodology we propose brings exciting contributions into epidemic surveillance and modeling. While voluntary, prevalence-based local awareness has been thoroughly studied in the modeling literature[12,15–17], there has been little empirical evidence about its impact in real epidemics. We provide such evidence through an innovative approach based on genetic sequence data, which we carefully validate in simulation experiments. Furthermore, the temporary drops in local awareness behavior – detected in both the genetic data and the survey experiment, raise important questions about the underlying mechanisms driving these fluctuations and how often they go unnoticed during pandemics. Our measurements also provide guidance for the design of future awareness models, shifting from intuition-based assumptions to insights derived from real-world data.

From an operational perspective, by studying MECS values, we are able to measure how effectively different countries managed to contain superspreading events in different waves. We observe that this effectiveness is highly correlated with the containment policies implemented in each country, suggesting that stricter government policies could motivate the public to undertake stricter voluntary prevention methods. We envision that similar analyses will be used to evaluate the effectiveness of the implemented policies in future pandemics, potentially generating a positive feedback loop between cooperative preventive behavior and epidemic containment. Unfortunately, even with the rapid advancement of genetic sequencing technologies, the financial burden of achieving the sequencing rate necessary for our proposed analysis is quite high, and we cannot expect that we will have the same coverage in every pandemic. Deciding how much sequencing is actually needed for epidemic surveillance is currently an active research topic, as the cost-benefit trade-offs are still being debated[54]. Our analysis adds to this discussion by bringing a new potential benefit of dense genetic sequencing.

Finally, we highlight the importance of continuing this research towards more specific questions, such as understanding the socio-economic factors that determine the outcome of superspreading events, and whether the measured local awareness behavior is externally-imposed or voluntary, as it was asked in the questionnaire in Fig. 1a. Large-scale genetic data analysis provides a new opportunity to answer these questions, and to further our understanding about the underlying mechanisms of behavior-disease models.

## Methods

### Datasets and preprocessing

**MASZK survey.** The MASZK telephone survey was collected over 26 months (between April 2020 and July 2022) from a nationally representative sample of 1000 respondents every month in Hungary via the Computer-Assisted Telephonic Interview (CATI) methodology[18]. The survey included standard questions on contact and vaccination behavior (not shown), as well as questions about the types of preventive behavior (mask wearing and social distancing shown in Fig. 1a) practiced by the respondent the day before the survey was taken.

During the last 9 months of data collection the following question on local awareness behavior was asked from the respondents: (translated, originally in Hungarian) "If several of your close contacts got infected, how likely are you to start taking better precautions against the coronavirus, either by wearing a mask more often or by reducing the number of people you meet? Please answer on a scale from 1 to 5, where 1 means that you would definitely not start taking better precautions in the given situation, and 5 means that you would definitely start taking better precautions." Fig. 1a shows the average and the confidence interval under the normality assumption of the scores collected from the respondents without further preprocessing.

**GISAID EpiCoV genetic database.** We downloaded the entire GISAID EpiCoV database between March 2020 and March 2023[29]. Although the database contains sequences from over 200 countries worldwide, we kept only European countries with sequencing rate at least 2% from the Delta, Omicron BA.1 and BA.2 variants, in the interest of data quality and to match the survey experiment. Our analysis mainly relies on the amino-acid-level substitution dataset of each sequence compared to the WIV04 reference sequence collected in late 2019 in Wuhan. Although the amino-acid-level substitution data is more aggregated than the raw genetic data (three nucleotides encode one amino acid, with multiple triplets having the same encoding), it still contains highly detailed information about the genetic code of the samples. We filtered out samples where the substitution data was not computed on the full-length virus genome. Besides the amino-acid substitutions, the dataset also contains various metadata, such as the date and the location of the sample (usually at the country or county level). The collision clusters were computed by binning the samples based on their substitution profile and country-variant pair. To rule out mass importations from abroad, we removed clusters that have at least 10 sequences in at least two countries.

### Superspreading event detection

Let $CC_{c,i}$ denote the size (number of samples) of the collision cluster at time $t$ (integer value measured in weeks), its country-variant pair denoted by $c$, and its cluster index $i$ (Fig. 2b). We track the normalized changes in collision cluster sizes defined as

$$\text{NormChange}_{c,i}(t) = \frac{CC_{c,i}(t+1) - CC_{c,i}(t)}{\max(1, \sqrt{CC_{c,i}(t)})}. \qquad (3)$$

The normalization with the square root of the collision cluster size accounts for the natural fluctuation of the cluster sizes. Indeed, assuming that the patients in the collision clusters at time $t$ independently infect an identically distributed random number of new patients with the same amino acid signature at time $t + 1$, by the Central Limit Theorem, we expect the fluctuations of $CC_{c,i}(t+1)$ to be proportional to the square root of $CC_{c,i}(t)$.

We say that a superspreading event happens at time $i$ in collision cluster $(c, i)$ if $\text{NormChange}_{c,i}(t)$ is larger than a threshold, which is set to 9 by default following[40], and we give a robustness analysis for this value in Supplementary Material B.1. With this definition, it is possible that one collision cluster contains multiple superspreading events, although we only observe this in very few cases in the real data. See Supplementary Material A for a detailed explanation of the methodological choices in this section, and additional validation steps based on the location metadata.

### ECS assignment

In each country-variant pair with at least 15 detected superspreading events, we match each superspreading event $(c, i, t)$ with at least $2m = 10$ baseline events (not superspreading events) based on collision cluster sizes (see Supplementary Material B.2 for a robustness analysis on the value of $m$). We outline a procedure that ensures that compared to $(c, i, t)$, at least $m$ larger and $m$ smaller collision clusters are always selected as baselines, however, if there is a large number of collision clusters with the same time as $(c, i, t)$, then we select all of them to avoid arbitrary selections and to make use of the available data.

Formally, let us denote the cluster indices (resp., time indices) of the matched collision clusters by $I(c, i, t)$ (resp., $T(c, i, t)$). First, we sort all baseline events that have size at least as large as the superspreading event detection threshold (9) by sampling time to create an order $\mathcal{O}$. We construct $I(c, i, t)$ (resp., $T(c, i, t)$)) by taking the union of the cluster (resp., time) indices of all collision clusters sampled at time $t$, as well as the $m$ closest previous and the $m$ closest subsequent collision clusters to $(c, i, t)$ in $\mathcal{O}$. Then, the median baseline NormChange values at time $t$

 

**Table 2 | The specific awareness functions implemented in our synthetic models**

| Name | Equation |
|------|----------|
| No awareness: | $\beta_{u,t} = \beta_0$ |
| Exponential local awareness (1): | $\beta_{u,t} = \beta_0 \cdot \exp(-\alpha_l I_{u,t})$ |
| Exponential global awareness (2): | $\beta_{u,t} = \beta_0 \cdot \exp(-\alpha_g I_t / N)$ |
| Linear local awareness: | $\beta_{u,t} = \beta_0 \cdot 1/(1 + \alpha_l I_{u,t})$ |
| Linear global awareness: | $\beta_{u,t} = \beta_0 \cdot 1/(1 + \alpha_g I_t / N))$, |

are defined as

$$\text{Baseline}_{c,i}(t) = \text{median}_j \left( \text{NormChange}_{c, I(c,i,t)_j}(T(c,i,t)_j) \right), \quad (4)$$

where the NormChange function is defined in equation (3). Thereafter, $\text{ECS}_{c,i}(t)$ is computed as

$$\text{ECS}_{c,i}(t) = \text{Baseline}_{c,i}(t+1) - \text{NormChange}_{c,i}(t+1). \quad (5)$$

and MECS for country $c$ is defined as the median of the $\text{ECS}_{c,i}(t)$ values for all superspreading events $(c, i, t)$ in $c$.

In Fig. 4 and Supplementary Fig. B3, MECS values are compared with various exogenous variables (sequencing rate, attack rate, Containment Health Index). These exogenous variables are computed for each country on a weekly basis based on publicly available datasets on the case counts[55] and the Oxford Containment Health Index[41]. Then, each superspreading event in the dataset is matched with the exogenous variables based on the time and country information. Finally, the plotted values are computed as the median of the exogenous variables of the superspreading events corresponding to index $c$ (which are also used to compute MECS). See Supplementary Material A for a detailed explanation of these methodological choices in this section.

### Generating synthetic networks
Geometric Inhomogeneous Random Graphs (GIRGs) were generated by sampling the spatial coordinates and the expected degrees of the nodes, and then connecting them by edges with a probability given by a kernel function, which is inversely proportional with the spatial distance, and assures the desired node degrees[48]. To sample networks with a heterogeneous degree distribution and geometric properties[56], we set the degree exponent to $\tau = 3.5$ and the parameters to $\alpha = 2.3$, $C_1 = 0.8$. We tuned $C_2$ numerically to achieve the desired average degree (by default 3). Configuration models are generated by degree-preserving edge shuffling of the edges of the generated GIRG networks. SBMs were generated with blocks of size 50. The connection probabilities inside and between the blocks were tuned so that for each node, half of its average degree was inside the block, and half of its average degree was outside the block. All synthetic networks had $10^4$ nodes, and we took the largest connected component if the network was not connected. We include a visualization of the size, degree distribution and average clustering coefficient of the generated networks in Supplementary Fig. B8.

### SIR model extended with local and global awareness
On both synthetic and real networks, we used our own implementation of the SIR model. We model local and global awareness by setting the infection probability of an infectious node $u$ to any other susceptible node $v$ at time $t$ to a function $\beta_{u,t}$. In case of local awareness, $\beta_{u,t}$ depends on on $I_{u,t}$, the number of infected neighbors of $u$ at $t$, and in case of global awareness, $\beta_{u,t}$ depends on $I_t$, the total number of infected nodes at time $t$. The specific awareness functions we implemented are shown in Table 2. The default values for the basic infection

probability $\beta_0$ and the recovery probability $\gamma$ were always 0.15 (see Supplementary Section B.3).

For each set of parameters, we simulated $n = 200$ epidemic processes with different random seeds. When the underlying network was synthetic, it was generated with the same random seed as the epidemic process prior to the simulation.

### Generating synthetic genetic sequences
Once the epidemic process has been simulated, we assign synthetic genetic sequences to each node of the infection tree using the Jukes-Cantor genetic substitution model[44], which is the simplest genetic substitution model we could select for our application. More concretely, we assign strings of size 10 consisting of the digits {0, 1, 2, 3} to each infected node using the following procedure. First, we assign a uniformly randomly chosen string to the root of the infection tree. Thereafter, for each edge of the infection tree, we sample each digit of the string of the parent node with probability $p_{mut} = 0.0375$, change it to a uniformly random new digit (among the other three digits), and assign the resulting string to the child node. These parameters assure that the non-synonymous mutation probability during a transmission event agrees with estimates from the literature. Indeed, it has been reported that the SARS-CoV-2 virus has on average one mutation in every 2 generations[57], which under natural selection would imply a non-synonymous mutation probability of $0.77 \cdot 0.5 = 0.38$, based on the ratio of the number of non-synonymous to synonymous sites in SARS-CoV-2's genome[58]. However, since SARS-CoV-2 was predominantly under purifying selection[58], the true non-synonymous mutation probability was lower in most cases, motivating our parameter choice for $p_{mut}$, which results in a mutation probability of $1 - (1 - 0.0375)^{10} \approx 0.32$ during a transmission event. Finally, we also note that our synthetic genetic sequences are much shorter than the COVID-19 genetic sequences for the sake of computational efficiency.

### Reporting summary
Further information on research design is available in the Nature Portfolio Reporting Summary linked to this article.

## Data availability
The MASZK survey data used to generate Fig. 1 is shared in Supplementary Data 1. The simulated data in Fig. 3 is available in Supplementary Data 2. All genome sequences and associated metadata are published in GISAID's EpiCoV database. To view the contributors of each individual sequence with details such as accession number, Virus name, Collection date, Originating Lab and Submitting Lab and the list of Authors, visit https://doi.org/10.55876/gis8.240404rn. An acknowledgment table for the genetic sequences can be found in Supplementary Data 4. For the reported COVID-19 case numbers, we used the "JHU CSSE COVID-19 Data" available at https://github.com/CSSEGISandData/COVID-19 applying a 7-day rolling average and outlier detection to ensure data consistency and reliability. An intermediate dataset containing the accession numbers of the sequences, the computed ECS values, and various additional metadata of the collision clusters corresponding to the detected superspreading events is shared in Supplementary Data 3.

## Code availability
The code for the genetic data generation and analysis pipeline shown in Fig. 1b is available via Code Ocean at https://doi.org/10.24433/CO.1189503.v1.

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

## Acknowledgements

We thank Eszter Ari, Andreas Bergthaler, and Tamás Stirling for their insightful comments and remarks. We also thank Júlia Koltai and Gergely Röst for their contribution to the MASZK survey data collection. We gratefully acknowledge all data contributors, i.e., the Authors and their Originating laboratories responsible for obtaining the specimens, and their submitting laboratories for generating the genetic sequence and metadata and sharing via the GISAID Initiative, on which this research is based. GÓ was primarily supported by the Swiss National Science Foundation, under grant number P500PT-211129, with additional funding from the Austrian Science Fund (FWF) Cluster of Excellence "Microbiomes drive planetary health" (10.55776/COE7). M.K. was supported by the CHIST-ERA project SAI: FWF I 5205-N; the SoBigData++ H2020-871042; SoBigData-PPP HORIZON-INFRA-2021-DEV-02 program under grant agreement No 101079043, and the National Laboratory for Health Security, Alfréd Rényi Institute, RRF-2.3.1-21-2022-00006.

## Author contributions

GÓ and MK conceptualized the research design. GÓ conducted the data analysis, performed the synthetic simulations, created the visualizations and wrote the first draft of the manuscript. MK acquired the survey data and supervised the research. GÓ and MK edited the final version of the manuscript.

## Competing interests

The authors declare no competing interests.
