## [Transparent Peer Review file · Nature Communications]

Epidemic-induced local awareness behavior inferred from surveys and genetic sequence data

Corresponding Author: Dr Gergely Ódor

Version 0:

Reviewer comments:

Reviewer #1

(Remarks to the Author)

This manuscript addresses the very interesting hypothesis that SARS-CoV-2 transmission is in-part driven by changes in individual's willingness to change their behavior based on cases in their social network. The authors test this hypothesis through the use of survey data and novel cluster-based methods that leverage the wealth of genomic surveillance data generated throughout the pandemic.

I am not sufficiently familiar with network analysis to comment on simulation studies used by the authors and will focus my critique on genomic epidemiology used throughout the study.

The impact of local awareness on pathogen transmission is of great interest to the field, and the ability to efficiently measure local awareness from genomic data would be an important advancement. However, there are significant concerns with the manuscript in its current form.

****Major concerns****

Page 1: It is not clear how local awareness was defined in the Hungarian survey. This is key as figure 1 is presented as motivation for the rest of the study. The MASKZ citation is predates this dataset and focuses on social contact rates. Was the survey updated to include a question about local awareness, was this metric calculated from the contact data?

Page 3: The assumption that sequences assigned the same variant name have the same fitness is not valid. All cases since January 2022 have been Omicron, and yet there have been several waves caused by antigenically distinct Omicron sub-variants. This is important to address as many of the results only hold for Omicron. BA.1 clusters that appear early are expected to grow faster than those that appear near the end of the first wave as immune the BA.1 grows and BA.2 (and subvariants) grow in frequency.

I am uncertain if each cluster contains at most 1 super spreading event and the containment score is calculated from the next time point or if multiple time points within the same cluster can be classified as super spreading events.

The methods seems to suggest that for a given super spreading event the baseline change in cluster size is calculated based on the size of the cluster, the variant name, the country of sampling, and the time since the cluster was first observed in that country. This seems to imply that super spreading events early in the Omicron wave, when transmission was at its height, could be compared with clusters from later in the wave when the epidemic had a negative growth rate. The baseline cluster growth rate should be compared to clusters at similar points in the epidemic curve.

In the supplemental material the authors state "In principle, it is possible that not one but multiple patients with the same amino acid signature caused independent and simultaneous SSEs, however, since this is an unlikely event ...". But this is not true. It is known that genetic data underestimates the number of introductions into a country, because, among other reasons, genetically similar lineages are counted as one introduction. This is particularly the true in the case of the initial Omicron wave in the UK, which was caused by many introductions from abroad. Are the county-variant labels robust to international transmission? Are there genetic clusters that span countries in the dataset? This could bias the results, as it

increased cluster size could result from importations not super spreading. Are the results robust if only genetic clusters that arise within a single country are included?

This may stem from my ignorance in network theory, but how do we know that super spreading events which are contained are contain through increased local awareness and not variants rapidly depleting the number of susceptible nodes in a graph neighborhood? By definition they must grow faster than the background and must deplete susceptibles faster.

The manuscript relies heavily on novel acronyms (SSE, AACCs, ECS) which ask a lot of the reader. It is recommended to use simpler, familiar language (super spreading events, clusters, containment scores) instead.

****Minor concerns****

The rate of evolution used in this study (1 substitution/ 2 weeks) represents the approximate rate of nucleotide substitution, not the rate of amino acid substitution. Many (if not the majority) of substitutions observed in SARS-CoV-2 do not change the amino acid sequence. The nonsynonymous substitution rate should be used in the simulation.

Are the results robust if the nucleotide sequence is used instead of the amino acid sequence. A trial of the could be done with a subset of the data (Omicron - BA.1) , if the GISAID dataset is too large to process.

The reasoning for normalizing by the square root of the cluster size should be included in the main text, and not just the supplemental.

What do the dashed lines in the Figure 1 represent?

In some cases GISAID reports finer location data than country (e.g. county in the US). If available this could be used as evidence to support the veracity of the proposed super spreader events.

Figure 4: How were the default epidemiological parameters chosen?

Figure 5: Is there a similar hypothesis for the apparent drop in the containment metric at the start of the Delta wave in the UK.

(Remarks on code availability)

I have not run the code, but it appears well documented. The readme provides enough information to rerun the analysis provided GISAID data is available (it can not be shared here). The authors provide a demo analysis to help readers understand their analyses.

Reviewer #2

(Remarks to the Author)

I would like to thank the editor for inviting me to review this interesting manuscript, Epidemic-induced local awareness behavior inferred from surveys and genetic sequence data, which develops a method to infer the awareness using large-scale genetic sequence data. This novel method takes advantage of genetic sequence datasets, such as GISAID, by defining Amino Acid Collision Clusters (AACCs), computing their normalized changes to identify Superspreading Events (SSEs), and calculating Event Containment Scores (ECS) as an indicator of the containment of SSEs, which serves as a proxy of local awareness. While this proposed method is potentially valuable for epidemic surveillance and policy evaluation, significant improvements in clarity, structure, and methodological justification are necessary.

Recommendation: Major revision.

Major issues:

1. The organization of the manuscript is confusing. I felt lost immediately when reading the Introduction and found it difficult to understand how these poorly defined terms are related to the research question addressed in the manuscript. I have the following suggestions:
 - (1) clearly define key terms and concepts early in the Introduction;
 - (2) add a brief paragraph at the end of Introduction to summarise the organization of the paper;
 - (3) a Data and analysis subsection, similar to the current subsection 1.1, can be added to the Methods, introducing the genetic sequence data and the survey data in details;
 - (4) a Method overview subsection can be very useful within the Results because the current subsections in Results mix methodologies, results, and interpretation;
 - (5) separate the results and discussion more clearly;
 - (6) provide a more detailed methodology section in the main text, moving some content from the supplement;
 - (7) include a dedicated section on validation and robustness checks in the main text.
2. Subsection 2.2 serves as a validation and justification of using ECS as the proxy of local awareness based on simulation study. It may serve a better job when it is before the current subsection 2.1 and after a Method overview subsection. Also, the current version mixes methodologies and results, which is confusing.
3. Clear definitions will be needed for clarifying local awareness and global awareness, and policy-induced behavior pattern.
4. Awareness, local awareness, and global awareness are used throughout the manuscript, without clear definitions and distinctions.

5. Am I correct that the authors consider the Hungarian survey reflects the local awareness and the government imposed preventive measures reflect the global awareness? If so, any justification for that?
6. In the simulation studies, the authors explicitly define local awareness and global awareness as functions of the infections in the neighborhood (ie connections to the node) and of the total infections in the population, respectively. If the governmental preventive measures reflect the global awareness, they should be consistent with the case reports. Is that correct?
7. The validation of ECS by the observation of similar drop shown in the Hungarian survey during the omicron wave in other European countries is insufficient to me. I notice that the authors didn't compute the ECS over time in Hungry. Is it because the lack of sufficient sequence sampling? Because it will be more straightforward to validate by comparing the changing patterns between them? For instance, plot the ECSs in Hungary and compare it to the survey, the reported cases, side by side (similar to Figure 5a and Figure 1a).

Minor issues:

1. Figure 1:

- (1) What is the mean local awareness? What is the unit for it?
- (2) It will be more straightforward if the prevalence is in %.
- (3) AACC is not introduced in the caption.
- (4) In panel b, the light blue parts are not necessary because the synthetic data are for validation but not for actual implementation.

2. Figure 3:

- (1) What do the dark green bar indicate?
- (2) In the main text, the sequencing rate is called sampling rate; please keep the terminology cohesive.

(Remarks on code availability)

Version 1:

Reviewer comments:

Reviewer #1

(Remarks to the Author)

I thank the authors for taking the time to address my and the other reviewer's concerns. The presentation is much improved, and the key findings and methods are clearer.

Unfortunately, I still have concerns around the presentation of the genomic data in the UK (5C), and its relationship to the survey in Hungry, which together seem to make up the main finding of the manuscript.

The authors show that their containment metrics are robust to a number of potential confounding factors (such as sequencing rate and attack rate) but there are other processes which could account for the decreased containment scores at the start of the Omicron wave. If this trend was driven by behavior, shouldn't we also see a decrease in containment of Delta clusters as well? For example, BA.1 and BA.2 show similar ESC near March of 2022. It seems more likely the difference between Omicron and Delta ESC is due to differences in the transmissibility of the VOCs and their importation dynamics. Not behavioral changes.

It is difficult to know without looking at data in Figure 5C, but the drop in ESC seen in during BA.1 in the UK could be an artifact of the rapid growth of the variant there. Previous work (Tsui et. al, 2023) has shown the Omicron wave in the UK was driven by a few very large, introductions. Samples from the same introduction are genetically similar because of their recent ancestry, and given the explosive growth of these introductions I would expect many clusters would have positive and large NormChanges for multiple weeks (qualifying them as super spreaders). When compared against baseline clusters that by definition were not growing rapidly this would result in a negative ESC. It is also known that later introductions were smaller. Because these lineages had time to diversify elsewhere, they would likely form distinct clusters from those driving the wave. These dynamics are not captured in the simulations but could result in large NormChanges for clusters in importations which arrived early and small ones for those that arrived during the wave.

Very minor:

Figure 5C: It is difficult to determine which point line up with the months on the x axis. Axis ticks would make this easier.

A GISAID acknowledgement table should be included for the sequences used in the study.

(Remarks on code availability)

I have looked through the code. Since the code requires GISAID data, it would be helpful to provide intermediate files of the cluster sizes, EPI ids in each cluster, and norm changes used to generate the figures.

Reviewer #2

(Remarks to the Author)

The authors have addressed all of my concerns.

(Remarks on code availability)

Version 2:

Reviewer comments:

Reviewer #1

(Remarks to the Author)

The authors have addressed my comments. Thank you for taking the time to do so!

(Remarks on code availability)

Reviewer #1 (Remarks to the Author):

This manuscript addresses the very interesting hypothesis that SARS-CoV-2 transmission is in-part driven by changes in individual's willingness to change their behavior based on cases in their social network. The authors test this hypothesis through the use of survey data and novel cluster-based methods that leverage the wealth of genomic surveillance data generated throughout the pandemic.

I am not sufficiently familiar with network analysis to comment on simulation studies used by the authors and will focus my critique on genomic epidemiology used throughout the study.

The impact of local awareness on pathogen transmission is of great interest to the field, and the ability to efficiently measure local awareness from genomic data would be an important advancement. However, there are significant concerns with the manuscript in its current form.

We are thankful for the Reviewer to recognise the importance of our research question and for her/his constructive comments. Please find our point-by-point answers to all your questions and suggestions below in blue. We feel that the clarity of the manuscript and evidence of the robustness of our findings have improved as a result of these changes.

****Major concerns****

Page 1: It is not clear how local awareness was defined in the Hungarian survey. This is key as figure 1 is presented as motivation for the rest of the study. The MASKZ citation is predates this dataset and focuses on social contact rates. Was the survey updated to include a question about local awareness, was this metric calculated from the contact data?

Thank you for highlighting that our definition about the computation of the awareness scores in Figure 1 was not precise. As response to this comment, and to help the readers, we now include a Datasets and preprocessing subsection in the Methods section to clarify the details regarding the Hungarian MASZK survey.

Page 3: The assumption that sequences assigned the same variant name have the same fitness is not valid. All cases since January 2022 have been Omicron, and yet there have been several waves caused by antigenically distinct Omicron sub-variants. This is important to address as many of the results only hold for Omicron. BA.1 clusters that appear early are expected to grow faster than those that appear near the end of the first wave as immune the BA.1 grows and BA.2 (and subvariants) grow in frequency.

Thank you for pointing this out! We now split the Omicron subvariants and we focus mainly on BA.1 and BA.2. Unfortunately, this change results in a loss of several data points, as we do not detect enough superspreading events in many countries, which were previously included in the

analysis (even after reducing our threshold for the minimum number of SSEs from 20 to 15). Nevertheless, our results for BA.1 and BA.2 still point in the same direction as before.

We now report our results for the Delta and the Omicron BA.1 as well as all of the Omicron variants (merged as before) in the main text, and we report the results on the Omicron BA.2 in the Supplementary Materials. In the interest of space, since most of our results (from the survey and the genetic sequence analysis) were on the Delta and the Omicron variant, we omitted the results on the pre-Delta variants.

I am uncertain if each cluster contains at most 1 super spreading event and the containment score is calculated from the next time point or if multiple time points within the same cluster can be classified as super spreading events.

Yes, it is possible that one cluster contains multiple superspreading events, although this is very rare. We now include a clarification on this question in the Methods section.

The methods seems to suggest that for a given super spreading event the baseline change in cluster size is calculated based on the size of the cluster, the variant name, the country of sampling, and the time since the cluster was first observed in that country. This seems to imply that super spreading events early in the Omicron wave, when transmission was at its height, could be compared with clusters from later in the wave when the epidemic had a negative growth rate. The baseline cluster growth rate should be compared to clusters at similar points in the epidemic curve.

Thank you for raising this concern! We changed our matching algorithm so that it now selects baseline events with similar sampling time as the superspreading event. To make sure that the baseline events are still comparable in size, we only consider baseline events that are at least as large as the superspreading event detection threshold (9).

During this adjustment we realized that one of the subplots in our final figure (Fig 5b) was not robust to the changes. Therefore, we omitted this subfigure and replaced it with two other figures to perform a similar, but more aggregated analysis. Consequently, there are also changes in our interpretation of the results, however, these do not alter the main message or the significance of our paper.

In the supplemental material the authors state "In principle, it is possible that not one but multiple patients with the same amino acid signature caused independent and simultaneous SSEs, however, since this is an unlikely event ...". But this is not true. It is known that genetic data underestimates the number of introductions into a country, because, among other reasons, genetically similar lineages are counted as one introduction. This is particularly the true in the case of the initial Omicron wave in the UK, which was caused by many introductions from abroad. Are the county-variant labels robust to international transmission? Are there genetic clusters that span countries in the dataset? This could bias the results, as it increased cluster size could result from importations not super spreading. Are the results robust if only genetic clusters that arise within a single country are included?

Thank you for calling our attention to this potential bias! To address this insightful comment, we added an additional pre-filtering step to remove clusters that have at least 10 sequences in at least two countries. We observed that this pre-filtering only removed a handful of clusters, and had little effect on our main results.

This may stem from my ignorance in network theory, but how do we know that super spreading events which are contained are contain through increased local awareness and not variants rapidly depleting the number of susceptible nodes in a graph neighborhood? By definition they must grow faster than the background and must deplete susceptibles faster.

We agree with the Reviewer that this is a very important point, thus we pay more attention to it in our revised manuscript. In Section 2.2, we show by simulations in a number of null models that we do not observe positive ECS values without local awareness, unless the sampling rate is very low and the ECS becomes very noisy. We also discuss that in case of an extremely transmissible disease, the ECS could theoretically be positive in clustered networks, due to the depletion of susceptibles. In the same paragraph we argue that while this may happen on a few occasions, we do not expect these extreme spreading patterns to be a general trend for SARS-CoV-2. As we state in the introduction, our hypothesis that quickly contained superspreading events are a sign of local awareness behavior may not hold in each individual case, but we do expect it to hold on average.

The manuscript relies heavily on novel acronyms (SSE, AACCs, ECS) which ask a lot of the reader. It is recommended to use simpler, familiar language (super spreading events, clusters, containment scores) instead.

Thank you for this comment! We almost completely eliminated the SSE and AACC acronyms, and also reduced the number of ECS acronyms.

****Minor concerns****

The rate of evolution used in this study (1 substitution/ 2 weeks) represents the approximate rate of nucleotide substitution, not the rate of amino acid substitution. Many (if not the majority) of substitutions observed in SARS-CoV-2 do not change the amino acid sequence. The nonsynonymous substitution rate should be used in the simulation.

Thank you for pointing this out. We now estimate the nonsynonymous substitution rate in Methods 4.6, and use this value in our simulations.

Are the results robust if the nucleotide sequence is used instead of the amino acid sequence. A trial of the could be done with a subset of the data (Omicron - BA.1) , if the GISAID dataset is too large to process.

We aligned all the Delta variant samples using the mafft package to the Wuhan-1 genome, then computed nucleotide substitutions (instead of amino acid changes) and executed our pipeline on this dataset. Overall, while we detect fewer superspreading events, and similar correlations, we do not have statistical power to regenerate Figure 3 on this dataset (figure below). Since we did not see major changes in the general trends of the results, and since we lack the computational resources to uncover the exact differences between the two datasets, we decided to stop this analysis at this point.

The reasoning for normalizing by the square root of the cluster size should be included in the main text, and not just the supplemental.

We now include the reasoning in Methods 4.2.

What do the dashed lines in the Figure 1 represent?

The Reviewer probably means Fig 2. The dashed lines mark the weeks when a new variant became dominant, and serve as a visual aid for matching subfigures (a) and (b). We now included the explanation in the figure caption.

In some cases GISAID reports finer location data than country (e.g. county in the US). If available this could be used as evidence to support the veracity of the proposed super spreader events.

This is an interesting idea. We included a subfigure and a paragraph in Supplementary Material A to provide additional validation for the superspreading event detection algorithm on the Belgian dataset, where the location metadata is available at the settlement level.

Figure 4: How were the default epidemiological parameters chosen?

We included a new subsection in the Supplementary Material to explain our choices for the default epidemiological parameters based on the COVID-19 literature. Since it is difficult to pin down the epidemiological parameters to match historical reports, our strategy is to show that our results are robust across a reasonable range of the parameter space. During the writing of this

section, we realized that our default beta parameter (0.3) was relatively high compared to reported values, and reduced it by a factor of 2 (to 0.15).

Figure 5: Is there a similar hypothesis for the apparent drop in the containment metric at the start of the Delta wave in the UK.

Although the results are slightly changed with the new matching algorithm, there is still a slight drop in the containment metric in November 2021 during the Delta wave in the UK. Interestingly, there is also a small drop in the awareness scores in the Hungarian survey experiment at around the same time. However, these drops are smaller compared to the Omicron BA.1 wave, with the confidence intervals intersecting in both datasets in case of the Delta wave. Due to the weakness of the signals, we do not expand on this matter in the main text beyond mentioning it.

Reviewer #1 (Remarks on code availability):

I have not run the code, but it appears well documented. The readme provides enough information to rerun the analysis provided GISAID data is available (it can not be shared here). The authors provide a demo analysis to help readers understand their analyses.

We thank the Reviewer to remark that our code and demo analysis are well developed to help the readers and the reproducibility of our results.

Reviewer #2 (Remarks to the Author):

I would like to thank the editor for inviting me to review this interesting manuscript, Epidemic-induced local awareness behavior inferred from surveys and genetic sequence data, which develops a method to infer the awareness using large-scale genetic sequence data. This novel method takes advantage of genetic sequence datasets, such as GISAID, by defining Amino Acid Collision Clusters (AACCs), computing their normalized changes to identify Superspreading Events (SSEs), and calculating Event Containment Scores (ECS) as an indicator of the containment of SSEs, which serves as a proxy of local awareness. While this proposed method is potentially valuable for epidemic surveillance and policy evaluation, significant improvements in clarity, structure, and methodological justification are necessary.

Recommendation: Major revision.

We are thankful for the Reviewer to recognise the importance of our research question and for her/his constructive comments. Please find our point-by-point answers to all your questions and suggestions below in blue. We feel that the clarity of the manuscript and evidence of the robustness of our findings have improved as a result of these changes.

Major issues:

1. The organization of the manuscript is confusing. I felt lost immediately when reading the Introduction and found it difficult to understand how these poorly defined terms are related to the research question addressed in the manuscript. I have the following suggestions:

- (1) clearly define key terms and concepts early in the Introduction;
- (2) add a brief paragraph at the end of Introduction to summarise the organization of the paper;
- (3) a Data and analysis subsection, similar to the current subsection 1.1, can be added to the Methods, introducing the genetic sequence data and the survey data in details;
- (4) a Method overview subsection can be very useful within the Results because the current subsections in Results mix methodologies, results, and interpretation;
- (5) separate the results and discussion more clearly;
- (6) provide a more detailed methodology section in the main text, moving some content from the supplement;
- (7) include a dedicated section on validation and robustness checks in the main text.

We sincerely thank the Reviewer for proposing a clear and well-organized structure for our paper. Following the numbering of the Reviewer, we included more precise definitions (1), a summary in the last paragraph of the Introduction (2), a Data and preprocessing subsection (3), and a method overview section (4). We also aimed at separating the results and the discussion more clearly (5) and moved some content regarding the normalization by the square root of the cluster sizes into the main text (6). These modifications all increased the word count of the main text, thus in the interest of space and avoiding repetitions, we kept the dedicated section on validation and robustness checks in the Supplementary Materials (7).

2. Subsection 2.2 serves as a validation and justification of using ECS as the proxy of local awareness based on simulation study. It may serve a better job when it is before the current subsection 2.1 and after a Method overview subsection. Also, the current version mixes methodologies and results, which is confusing.

Thank you for the suggestion, we moved subsection 2.2. in between the method overview and the results on real data, as the Reviewer suggested.

3. Clear definitions will be needed for clarifying local awareness and global awareness, and policy-induced behavior pattern.

Thank you for pointing this out. We rewrote the second paragraph in the Introduction to clearly define local and global awareness as changes in preventive behavior induced by local and global information, respectively. We also make a distinction between voluntary and externally-imposed local awareness behavior, and explain which version we measure in which dataset. We do not distinguish between voluntary and externally imposed global awareness behavior, because global awareness is not a focus of the paper; it mainly appears as a confounding factor.

4. Awareness, local awareness, and global awareness are used throughout the manuscript, without clear definitions and distinctions.

Thank you again for the warning. We now carefully specify whether we are talking about local or global awareness in the text.

5. Am I correct that the authors consider the Hungarian survey reflects the local awareness and the government imposed preventive measures reflect the global awareness? If so, any justification for that?

Yes, we do consider the Hungarian survey as a proxy for local awareness. The reason is that we explicitly asked the respondents whether they would change their preventive behavior if the prevalence of the disease would increase among their close contacts – the definition of voluntary, prevalence-based local awareness.

In contrast, global awareness is defined by behavioral changes due to public information. Government imposed preventive measures are examples of public information sources that could induce global awareness, but we do not claim that they are reliable proxies at all times.

In this paper, we do not focus on global awareness, we just aim to rule it out as a potential confounding factor when monitoring local awareness. We achieved this primarily by including previously proposed global awareness models in case of the simulation results, and showing that they are not able to produce positive ECS values.

6. In the simulation studies, the authors explicitly define local awareness and global awareness as functions of the infections in the neighborhood (i.e. connections to the node) and of the total infections in the population, respectively.

If the governmental preventive measures reflect the global awareness, they should be consistent with the case reports. Is that correct?

The Reviewer is correct to expect a negative correlation between the governmental preventive measures and the case reports (at least in the early COVID waves). However, in this paper, we do not focus on global awareness, we just aim to rule it out as a potential confounding factor when monitoring local awareness. Therefore, it is out of our scope to provide further evidence to support the hypothesis of the Reviewer.

7. The validation of ECS by the observation of similar drop shown in the Hungarian survey during the omicron wave in other European countries is insufficient to me. I notice that the authors didn't compute the ECS over time in Hungary. Is it because the lack of sufficient sequence sampling? Because it will be more straightforward to validate by comparing the changing patterns between them? For instance, plot the ECSs in Hungary and compare it to the survey, the reported cases, side by side (similar to Figure 5a and Figure 1a).

We agree with the Reviewer, that it would be ideal to compute the ECSs for Hungary, and match them with the survey results. Unfortunately, even to this date, only about 1000 sequences were collected/shared in Hungary, and our proposed method is not able to operate with such a low

sequencing rate (either the data is very biased, or we do not detect any superspreading events). We now comment on this fact in the main text.

Minor issues:

1. Figure 1:

(1) What is the mean local awareness? What is the unit for it?

Thank you for letting us know that we were not precise about how we compute the awareness scores in Figure 1. We now include a Datasets and preprocessing subsection in the Methods section to clarify the details regarding the Hungarian MASZK survey.

(2) It will be more straightforward if the prevalence is in %.

Thank you for pointing this out, we changed the figure so that the prevalence and the preventive measures are all on the same scale.

(3) AACCC is not introduced in the caption.

Thank you for pointing this out! We removed the AACCC acronym for better readability.

(4) In panel b, the light blue parts are not necessary because the synthetic data are for validation but not for actual implementation.

We agree that the previous version of panel b was more confusing than informative. We changed the color coding of the figure so that light blue shows the synthetic validation pipeline (which we believe is still useful for the understanding of the paper) and grey shows the real data pipeline.

2. Figure 3:

(1) What do the dark green bar indicate?

Previously, in Figure 3 (now Figure 4) we plotted the ECS with green bars and the various exogenous variables with blue bars, as described in the figure caption. When the two bars were on top of each other, they were shown with dark green.

We realized that this was a confusing way to show these two variables and changed the figure to bar plots that are half the width. We hope that this version is more interpretable.

(2) In the main text, the sequencing rate is called sampling rate; please keep the terminology cohesive.

Thank you for bringing this to our attention. We fixed the inconsistencies in our terminology regarding the sequencing rate.

Reviewer #1 (Remarks to the Author):

I thank the authors for taking the time to address my and the other reviewer's concerns. The presentation is much improved, and the key findings and methods are clearer.

Thank you for the encouraging words and for carefully reading our manuscript for the second time.

Unfortunately, I still have concerns around the presentation of the genomic data in the UK (5C), and its relationship to the survey in Hungry, which together seem to make up the main finding of the manuscript.

The authors show that their containment metrics are robust to a number of potential confounding factors (such as sequencing rate and attack rate) but there are other processes which could account for the decreased containment scores at the start of the Omicron wave. If this trend was driven by behavior, shouldn't we also see a decrease in containment of Delta clusters as well? For example, BA.1 and BA.2 show similar ESC near March of 2022. It seems more likely the difference between Omicron and Delta ESC is due to differences in the transmissibility of the VOCs and their importation dynamics. Not behavioral changes.

We have carefully considered the issue raised by the Reviewer, however, we continue to believe that a behavioral change is the most plausible explanation for our results. With regards to the discontinuity at the start of the BA.1 wave, we may simply lack the temporal resolution to capture the change in behavior between the two waves. We now compute the ECS values on a bi-weekly basis via a month-long rolling window, and we observe that the ends of the curves are closer together, but there is a gap where we do not get a stable signal. We now explain this in the main text.

We also recognize that we are treating the entire UK as one population, whereas during the introduction of the Omicron wave, the disease prevalence was far from homogenous in the country. Indeed, based on the reference [Tsui et. al, 2023] (we thank the Reviewer for bringing this paper to our attention) we now discuss the possibility that the drop in ECS may have been a result of an extreme overrepresentation of the capital city among the samples. We acknowledge that this makes the comparison between the ECS and the representative surveys even more difficult, however, even in this scenario, we are measuring a difference in behavior.

The reason that we do not think that the drop in ECS is due to an increase in transmissibility is because this would contradict our simulation results. Indeed, in Figure 3 we show that the ECS does not significantly depend on the parameter beta. This independence is due to our normalization – if the transmissibility is higher than the baseline NormChanges are also higher.

Regarding the start of the BA.2 wave, this is a more ideal scenario, because there is no temporal gap between the final BA.1 and the earliest BA.2 datapoint. We see an agreement in

the ECS values, despite BA.2 being more transmissible – again supporting the behavioral hypothesis.

It is difficult to know without looking at data in Figure 5C, but the drop in ESC seen in during BA.1 in the UK could be an artifact of the rapid growth of the variant there. Previous work (Tsui et. al, 2023) has shown the Omicron wave in the UK was driven by a few very large, introductions. Samples from the same introduction are genetically similar because of their recent ancestry, and given the explosive growth of these introductions I would expect many clusters would have positive and large NormChanges for multiple weeks (qualifying them as super spreaders). When compared against baseline clusters that by definition were not growing rapidly this would result in a negative ESC. It is also known that later introductions were smaller. Because these lineages had time to diversify elsewhere, they would likely form distinct clusters from those driving the wave. These dynamics are not captured in the simulations but could result in large NormChanges for clusters in importations which arrived early and small ones for those that arrived during the wave.

The Reviewer is correct to expect that NormChanges are larger early in an epidemic wave and smaller later on. However, this discrepancy is taken into account by the fact that we are matching baseline clusters based on time. The early superspreading events are matched with early baseline events, and in principle have the same potential to have large NormChanges in the next timestep (ignoring behavioral changes). Similarly, superspreading events in later introductions were compared to baseline events at the same time.

While we do not take introductions into account, our simulations do capture the asymmetry of low genetic diversity early in the wave and large diversity later in the wave. We show this using the plot below, where we observe (both with or without awareness) that the entropy of the cluster size distribution starts out low and remains high for some time during the descending part of the epidemic wave.

Finally, we also point out that we do not see the same drop in ECS in the BA.1 wave compared to the BA.2 wave in other countries (such as Denmark, Germany or Slovenia). This again hints

at a difference in behavior and not the transmissibility of the disease. We explain these differences by the observation that the Omicron wave arrived a few weeks earlier to the UK, and had the opportunity to grow.

Very minor:

Figure 5C: It is difficult to determine which point line up with the months on the x axis. Axis ticks would make this easier.

Thank you for noticing that the axis ticks were accidentally hidden. We fixed this issue.

A GISAID acknowledgement table should be included for the sequences used in the study.

Thank you for noticing that we forgot to submit the Supplemental Table sent by GISAID when we created our EPI_SET ID. We have now included it with the submission.

Reviewer #1 (Remarks on code availability):

I have looked through the code. Since the code requires GISAID data, it would be helpful to provide intermediate files of the cluster sizes, EPI ids in each cluster, and norm changes used to generate the figures.

Thank you for the suggestion and for looking through our code. We created a new folder called "intermediate_files", which contains the necessary files to reproduce Figures 1 and 3-5. We also include a new jupyter notebook ("plots/reproduce_figures_from_intermediate.ipynb") where the figures are reproduced. Among intermediate files (which we now also submit with the manuscript as supplementary data), "figure4-5_cluster_info.csv" contains the superspreading event sizes, the NormChange values, the ECS values and the EPI IDs.

Reviewer #2 (Remarks to the Author):

The authors have addressed all of my concerns.

Thank you for checking our manuscript for the second time. We are glad that we were able to improve based on the Reviewer's previous comments.